# Association of CX36 Protein Encoding Gene *GJD2* with Refractive Errors

**DOI:** 10.3390/genes13071166

**Published:** 2022-06-28

**Authors:** Edita Kunceviciene, Tomas Muskieta, Margarita Sriubiene, Rasa Liutkeviciene, Alina Smalinskiene, Ingrida Grabauskyte, Ruta Insodaite, Dovile Juoceviciute, Laimutis Kucinskas

**Affiliations:** 1Institute of Biology Systems and Genetic Research, Lithuanian University of Health Sciences, Eiveniu 4, 50161 Kaunas, Lithuania; tomas.muskieta@lsmu.lt (T.M.); margarita.sriubiene@lsmuni.lt (M.S.); alina.smalinskiene@lsmuni.lt (A.S.); ruta.insodaite@lsmu.lt (R.I.); dovile.juoceviciute@stud.lsmu.lt (D.J.); laimutis.kucinskas@lsmuni.lt (L.K.); 2The Institute of Cardiology, Lithuanian University of Health Sciences, Sukileliu 17, 50157 Kaunas, Lithuania; 3Department of Ophthalmology, Lithuanian University of Health Sciences, Eiveniu 2, 50161 Kaunas, Lithuania; rasa.liukeviciene@lsmu.lt; 4Neuroscience Institute, Lithuanian University of Health Sciences, Eiveniu 4, 50161 Kaunas, Lithuania; 5Department of Physics, Mathematics and Biophysics, Lithuanian University of Health Sciences, Eiveniu 4, 50161 Kaunas, Lithuania; ingrida.grabauskyte@lsmuni.lt

**Keywords:** *GJD2*, *RASGRF1*, refractive errors, SNPs, hyperopia, myopia, astigmatism

## Abstract

Purpose: This study aimed to evaluate the associations of *GJD2* (rs634990, rs524952) and RASGRF1 (rs8027411, rs4778879, rs28412916) gene polymorphisms with refractive errors. Methods: The study included 373 subjects with refractive errors (48 myopia, 239 myopia with astigmatism, 14 hyperopia, and 72 hyperopia with astigmatism patients) and 104 ophthalmologically healthy subjects in the control group. A quantitative real-time polymerase chain reaction (qPCR) method was chosen for genotyping. Statistical calculations and analysis of results were performed with IBM SPSS Statistics 27 software. Results: The correlations in monozygotic (MZ) twin pairs were higher compared to DZ pairs, indicating genetic effects on hyperopia and astigmatism. The heritability (h^2^) of hyperopia and astigmatism was 0.654 for the right eye and 0.492 for the left eye. The *GJD2* rs634990 TT genotype increased the incidence of hyperopia with astigmatism by 2.4-fold and the CT genotype decreased the incidence of hyperopia with astigmatism by 0.51-fold (*p* < 0.05). The *GJD2* rs524952 AT genotype reduced the incidence of hyperopia with astigmatism by 0.53-fold (*p* < 0.05). Haplotype analysis of SNPs in the *GJD2* gene revealed two statistically significant haplotypes: ACTAGG for rs634990 and TTTAGA for rs524952, which statistically significantly reduced the incidence of hyperopia and hyperopia with astigmatism by 0.41-fold (95% CI: 0.220–0.765) and 0.383-fold (95% CI: 0.199–0.737), respectively (*p* < 0.05). It was also found that, in the presence of haplotypes ACTAGG for rs634990 and TATAGA for rs524952, the possibility of hyperopia was reduced by 0.4-fold (*p* < 0.05). Conclusions: the heritability of hyperopia and hyperopia with astigmatism was 0.654–0.492, according to different eyes in patients between 20 and 40 years. The GJD2 rs634990 was identified as an SNP, which has significant associations with the co-occurrence of hyperopia and astigmatism. Patients with the *GJD2* gene rs634990 TT genotype were found to have a 2.4-fold higher risk of develop hyperopia with astigmatism.

## 1. Introduction

Refractive errors occur when the refractive system of the eye is unable to refract the light rays properly, so the light rays do not focus on the retina [1,2,3]. Refractive errors reflect a mismatch between the axial length of the eye and its optical power, resulting in blurred retinal images [4]. Refractive conditions of the eye are divided into emmetropia, myopia (near-sightedness), and hyperopia (far-sightedness). Emmetropia is described as a state of refraction in which light rays are perfectly focused into the pit of the macula in the retina. Myopia occurs when light rays focus in front of the retina. In hyperopia, the eye focuses rays from distant objects at a theoretical point behind the retina. Astigmatism refers to the different focusing of light passing through different meridians of the cornea [1,2,3,5]. Refractive errors are the leading cause of mild to moderate visual impairment, and knowledge of the prevalence of refractive error and risk factors provides a reliable basis for the development of future preventive interventions [5,6]. Researchers report that refractive errors begin with the interaction of complex genetic and environmental factors [6,7]. A causal link between increased years of education and more myopia cases has been confirmed by Mendelian randomization, and a significant association between time spent outdoors and less myopia was also observed [5].

Castagno et al. conducted a review that included 40 cross-sectional studies of hyperopia-independent populations. The prevalence of hyperopia ranged from 8.4% at six years of age, to 2–3% from 9 to 14 years of age, and approximately 1% at 15 years of age. The prevalence of hyperopia is higher in white children and those who live in rural areas. There is no consensus on the association between hyperopia and gender, family income, and parental schooling [8].

The first genome-wide association studies (GWAS) have been successfully used to understand the genetic background of refractive errors. These studies helped identify two significant loci (15q14 and 15q25) near the gap junction protein delta-2 (*GJD2*) gene and *RASGRF1* gene responsible for the RAS signaling pathway, which is associated with refractive errors. The *GJD2* gene encodes the protein connexin 36 (Cx36). Cx36 forms intermediate junction channels between adjacent neuronal cells found in photoreceptors, amacrine, and bipolar cells. In addition, Cx36 plays an important role in the transmission of visual signals and is involved in the transport of molecules and ions. The *RASGFR1* gene is important in regulating the RAS signaling pathway involved in synaptic transmission in photoreceptor cells. A particularly high expression of the *RASGRF1* gene is found in the retina and neurons [6].

Quint et al. found that depletion of gjd2a (Cx35.5) or gjd2b (Cx35.1) orthologs in zebrafish caused changes in the biometry and refractive status of the eye. ScRNA sequencing studies showed that Cx35.5 (gjd2a) is a retinal connexin, and its depletion leads to hyperopia and electrophysiological changes in the retina [9].

Refractive errors are common ocular disorders, characterized by a mismatch between focal power and axial length [10]. CX36 plays an important role in the transmission process in the retinal electrical circuit, between photoreceptor, amacrine, and bipolar cells, forming an intracellular transport of small molecules and ions [11]. *GJD2* (Cx36)-dependent rod pathways contribute to the release of dopamine from dopaminergic amacrine cells through the input of excited ON cone bipolar cells [12]. Mutations in the *GJD2* gene affect connexin activity, which affects the formation of the gap junction [13]. Regarding the association of rod pathways with the dopaminergic system, the lack of rod pathway negatively affects the number of dopaminergic amacrine cells and retinal dopamine/3,4-dihydroxyphenylacetic levels [14]. It is known from previous publications that dopamine receptors in the eye are involved in the development of vision-induced eye growth and refraction [15,16].

Based on the results in the gjd2a (Cx35.5) mutant fish, researchers hypothesized that the uncoupling of retinal gap junctions inhibits ocular growth. There is also evidence that pharmacological uncoupling of gap junctions using meclofenamic acid protected against form-deprivation myopia (FDM), an experimental form of myopia, in chicks [17,18]. Therefore, it can be predicted that, in the future, drug therapy may be applied to people with refractive errors. For now, it is important to find molecular markers present in people with refractive errors and who have a change in the gene sequence critical to protein quality. We investigated the single nucleotide polymorphisms (SNPs) of the *GJD2* and *RASGRF1* genes, to explain the significance of these genes for refractive errors by dividing refractive disorders into myopia, hyperopia, and astigmatism. Researchers have focused on myopia and *GJD2* gene molecular research; however, we were interested in evaluating the links between *GJD2* and hyperopia with astigmatism.

This study is important for the identification of gene variants responsible for hyperopia and astigmatism in humans, in cases of hyperopia with astigmatism and emmetropia. Comprehensive knowledge of the mechanisms of gene expression, including human studies with special cells and animal models, would help to discover pharmacogenetic substances that can inhibit/activate gene expression. We believe that with sufficient knowledge of the target genes and their mechanisms, this knowledge will be applied for the development of drugs to help the public prevent or reduce the occurrence of refractive errors.

Previous GWAS studies have identified two hyperopia important loci, 15q14 (rs11073060) and 8q12 (rs10089517) [19]. We performed an analysis of five SNPs found at loci 15q14 (rs634990, rsrs524952) and 15q25 (rs8027411, rs4778879, rs28412916), to assess their association with refractive errors.

## 2. Materials and Methods

### 2.1. Ethics Statement

Permission (Number BE-2-41) to undertake the study was obtained from Kaunas Regional Biomedical Research Ethics Committee. The procedure and purpose of the study were explained, and informed consent was obtained from all participants.

### 2.2. Study Samples

Study subjects with the diagnosed ocular refractive error were divided into 4 groups. Group 1 included subjects with mild (−0.5 to −2.9 D), moderate (−3 to −5.9 D), or high (<−5.99 D) myopia, and this group consisted of 48 subjects. Group 2 included individuals who had myopia with astigmatism; this group consisted of 239 subjects. Group 3 included low-grade (from +0.5 to +2.24 D), moderate (from +2.25 to +5.24 D), and high hyperopia (>+5.25 D), in a total of 14 individuals. Group 4 subjects had hyperopia with astigmatism; this group consisted of 72 subjects. Group 5 was a control group of healthy individuals. Exclusion criteria for the study group: (1) cataract; (2) previous interventions that may have affected refraction; (3) age ≤ 18 and ≥40; (4) refusal to participate in the investigation. Criteria for inclusion in the study group: (1) individuals with full ophthalmology; (2) age: ≥18 and ≤40.

### 2.3. Refractive Error Measurement

Refractive error was measured by an autorefractor (Accuref-K9001, Shin-Nippon, Japan) after solution. Cyclopentolate 1% was administered, and the mean spherical equivalent was calculated for each eye of every individual. The mean spherical equivalent was calculated using the standard formula: spherical equivalent = sphere + (cylinder/2).

### 2.4. DNA Extraction

Peripheral blood samples were collected from all individuals in ethylenediaminetetraacetic (EDTA) tubes for DNA extraction. DNA was extracted from leukocytes using a reagent kit (NucleoSpin Blood L Kit; Macherey & Nagel, Düren, Germany). DNA samples from one member of each MZ pair were used for genotyping.

### 2.5. Genotyping

The genotyping of two SNPs of the *GJD2* (rs634990, rs524952) and three SNPs of the *RASGRF1* (rs8027411, rs4778879, rs28412916) were performed for 477 individuals.

An Applied Biosystems 7900HT Real-Time Polymerase Chain Reaction System was used for detecting the SNPs. The cycling process started with heating for 10 min at 95 °C, followed by 40 cycles of 15 s at 95 °C and 1 min at 60 °C. Allelic discrimination was carried out using the software of Applied Biosystems.

### 2.6. Verification of Zygosity

Zygosity was determined using a DNA test. A polymerase chain reaction set (AmpFlSTR^®^ Identifiler^®^, Applied Biosystems, Foster City, CA, USA) was used to amplify short tandem repeats; 15 specific DNA markers were used for the comparison of genetic profiles: D8S1179, D21S11, D7S820, CSF180, D3S1358, TH01, D13S317, D16S539, D2S1338, D19S433, vWA, TROX, D18S51, D5S818, and Amelogenin.

### 2.7. Twin Method

The heritability of hyperopia and astigmatism was assessed using the twin method. The narrow-sense heritability coefficient (*h*^2^) was calculated in the interclass MZ and dizygotic (DZ) groups, based on Pearson correlations (r), according to the formula: *h*^2^ = 2 × (r_MZ_-r_DZ_). Heritability indicates the extent to which additive genes contribute to hyperopia and astigmatism

### 2.8. Statistical Analysis

First, we calculated descriptive statistics of the variables. The quantitative variables were described as median, minimum, and maximum, because variable distributions did not satisfy the normality assumption (Kolmogorov–Smirnov or Shapiro–Wilk tests). A nonparametric Mann–Whitney U test was used to determine differences in the distributions of continuous variables between two independent samples when the distribution of variables did not satisfy the normality assumption. In addition, we calculated the Spearman rank-order correlation coefficient between the continuous variables (distribution of variables did not satisfy the normality assumption). The qualitative variables were described using frequencies and percentages. A Chi-squared test was used to determine differences in categorical variables between the groups. A binary logistic regression was used to calculate the odds ratio for the associations between *GJD2* and *RASGRF1* gene variants and refractive errors.

A probability level of *p* < 0.05 was considered statistically significant. The program IBM SPSS Statistics 27 was selected for data processing.

## 3. Results

### 3.1. Sample Characteristics for Heritability Assessment of Hyperopia and Astigmatism

A total of 22 MZ and 21 DZ twin pairs participated in the heredity assessment studies. Subject age ranged from 20 to 40 years. The spherical equivalent of both axes ranged from 0.725 to 6.500 D (Table 1). No significant differences were found between the sex, age, and spherical equivalent of MZ and DZ twins.

### 3.2. Heritability of Hyperopia and Astigmatism using the Twin Method

Refractive errors for twin 1 versus twin 2 for the mean of spherical equivalent are shown in Figure 1 and Figure 2. Intrapair correlations for spherical equivalent were significantly higher in MZ twin pairs r = 0.389 (*p* = 0.013, 95% CI: 0.078–0.630) than in DZ twin pairs r = −0.213 (*p* = 0.164, 95% CI: −0.487–0.098) in the hyperopia and astigmatism group. The correlations in MZ pairs were clearly higher compared to DZ pairs, indicating genetic effects on hyperopia and astigmatism. For spherical equivalent, the heritability (*h*^2^) for hyperopia and astigmatism in the right eye was 0.654 and in the left eye 0.492.

### 3.3. Sample Characteristics for SNPs Studies

A total of 477 individuals from Lithuania participated in the study, of whom 185 were men and 293 were women. 373 subjects had ocular refractive errors and 104 were in the control group (healthy subjects). Subject age ranged from 18 to 40 years (Table 2). The mean age of the subjects was 23 years. The spherical equivalents (SE) of both eyes of the subjects ranged from −17.375 D to 5.125 D. In all the groups of refractive errors, the characteristics of SE differed significantly from the control group (all *p* < 0.05).

### 3.4. Associations of SNPs with Refractive Errors

The SNPs of two genes were examined for the association of gene variants with refractive errors: *GJD2* (rs634990, rs524952) and *RASGRF1* (rs8027411, rs4778879, rs28412916). The genotype distribution of the *GJD2* (rs634990, rs524952) and *RASGRF1* (rs4778879, rs28412916) polymorphisms was consistent with the Hardy–Weinberg equilibrium (HWE), and the *RASGRF1* rs8027411 polymorphism was not.

All possible genotypes were compared between the refractive error groups and the control group, and the possibility of having a refractive error was observed.

The *GJD2* rs634990 TT genotype was found to increase the risk of hyperopia with astigmatism by 2.4 times (95% CI: 1.146–4.860; *p* < 0.05) compared to the CT and CC genotypes. However, the CT genotype of rs634990 decreased the risk of hyperopia with astigmatism by 0.51 times (95% CI: 0.273–0.946; *p* < 0.05) compared with TT and CC genotypes. Studies of *GJD2* rs524952 gene variants showed that the AT genotype reduced the risk of hyperopia with astigmatism by 0.53 times (95% CI: 0.888–3.907; *p* < 0.05) compared to TT and AA genotypes (Table 3). No statistically significant differences were observed for other refractive errors and *RASGRF1* gene variants (*p* > 0.05).

### 3.5. Haplotype Analysis

Haplotype analysis of SNPs in the *GJD2* gene revealed two statistically significant haplotypes: ACTAGG for rs634990 and TTTAGA for rs524952, which statistically significantly reduced the incidence of hyperopia and hyperopia with astigmatism 0.410 (95% CI: 0.220–0.765) and 0.383 (95% CI: 0.199–0.737), respectively, (*p* < 0.05) (Table 3). It was also found that in the presence of haplotypes ACTAGG for rs634990 and TATAGA for rs524952, the possibility of hyperopia was reduced by about 0.4 times (*p* < 0.05) (Table 4 and Table 5).

### 3.6. Associations of SNPs with Degrees of Myopia and Hyperopia

The subjects were divided into two joint study groups: the first group had myopia and myopia with astigmatism (*n* = 287); the second group was hyperopia and hyperopia with astigmatism (*n* = 86). We evaluated the associations of *SNPs* with mild (from −0.5 to −2.9 D), moderate (from −0.9 to −2.9 D), and high (−6 D and less) myopia, and with low (from 0.5 to 2.24 D) and moderate hyperopia (from 2.25 to 5.24 D).

Studies of *GJD2* rs524952 gene variants showed that the AT genotype reduced the incidence of general low hyperopia in the right eye 0.26 times (95% CI: 0.067–0.971; *p* < 0.05) compared to AA and TT genotypes (Table 6). There were no significant differences between the remaining gene variants and the degree of refractive error in any of the studied groups (*p* > 0.05).

## 4. Discussion

Having had the opportunity to study twins, we found that the heritability of hyperopia and hyperopia with astigmatism in Lithuania is 0.654–0.492 (right and left eyes, respectively) in individuals between 20 and 40 years.

A previous publication found that the heritability of hyperopia is 0.75, with individuals over 40 years of age being assessed [4]. Researchers have found that astigmatism has a strong genetic basis, with astigmatism accounting for 50–60% of the predominant genetic effects compared to additive, common, or unique components [20]. In one of our publications, it was announced that the heritability of myopia in Lithuania is 0.66 [21]. However, heritability studies do not determine which genes are relevant to the onset of refractive errors. We performed *GJD2* and *RASGRF1* gene polymorphism studies in individuals with refractive errors and compared them with the control group.

*GJD2*-encoded CX36 protein-myopia relation studies are currently receiving a great deal of attention.

Based on the results with the gjd2a (Cx35.5) mutant fish, scientists think that the uncoupling of retinal gap junctions inhibits ocular growth, and that pharmacological uncoupling of gap junctions using meclofenamic acid protects against form-deprivation myopia [9,10].

In animal model studies, a significant reduction of *GJD2* mRNA and Cx36 protein expression in myopic eye retinal tissue was observed compared to control eyes [22,23]. This evidence directs more attention to explaining the development of refraction and ultimately the manifestation of myopia [24,25]. *GJD2* is thought to contribute to the genesis of myopia, by affecting retinal neuronal signaling [22].

Cx36, a distinct interstitial protein in the retina, is widely expressed in cones, AII amacrine cells, bipolar cells, and photoreceptor cells; therefore, any abnormality in the retinal tract may interfere with the normal development of refraction. There may also be an imbalance in the on and off signaling pathway caused by Cx36 dysfunction, which may affect the development of refraction. During normal image processing, the ON path cross-inhibits the OFF path to improve contrast coding efficiency [10,24].

In our previous studies, we investigated the association of these genes with myopia and found significant associations with combinations of *GJD2* (rs634990) and *RASGRF1* (rs8027411) genotypes [22]. However, by increasing the sample size and including other ocular refractive errors, as well as by exploring more SNPs, we found that the TT genotype increased the chance of hyperopia and astigmatism by 2.4 times compared to CT and CC genotypes. The CT genotype alone reduced the incidence of hyperopia with astigmatism 0.51 times compared to CC and TT genotypes (*p* < 0.05). The results obtained in this study are important, as they are among the first to show a positive association between *GJD2* rs634990 and hyperopia with astigmatism.

We also found that the haplotypes C-T and C-A of SNPs of the *GJD2* gene (rs634990, rs524952), respectively, were associated with a lower risk of hyperopia and hyperopia with astigmatism. However, we did not identify any significant haplotypes that would increase the incidence of hyperopia or hyperopia with astigmatism. Binary logistic regression with the *RASGRF1* gene rs8027411, rs4778879, and rs28412916 gene variants did not yield any statistically significant data with any of the comparative refractive error groups. The *RASGRF1* gene is thought to be associated with high-grade myopia [25,26], and we also believed that high-grade myopia should be significant in the results. Individuals with myopia from (−17.35; −1.25 D) participated in our study, but only 5.2% of the sample with myopia had a high degree of myopia (−6; >−6 D).

Therefore, we share the view of other researchers, that changes in the *GJD2* gene sequence may affect not only the development of myopia, but also farsightedness refraction by affecting neuronal signaling in the retina.

The depletion of gjd2a (Cx35.5) or gjd2b (Cx35.1) orthologs in zebrafish has been found to cause changes in ocular biometrics and refractive state. Single-cell RNA sequencing studies have shown that Cx35.5 (gjd2a) is a retinal compound, and its depletion causes hyperopia and electrophysiological changes in the retina [9].

Perhaps, further studies of *GJD2* gene interactions could not only be related to myopia but also consider cases of hyperopia. We also did not find any data on the pharmacological use of meclofenamic acid, and whether it may correct the refractive error in hyperopia.

To date, the known genes and candidate genes associated with astigmatism are *PDGFRA*, *CLDN7*, *ACP2*, and *TNFAIP8L*, of which *PDGFRA* has been shown to increase the risk of astigmatism 1.12 times [27]. Simson et al., in 2014, performed GWAS with refractive errors and found significant associations of *GJD2* rs524952 with myopia and hyperopia [19]. In this study, another *GJD2* gene polymorphism (rs634990) was identified that also showed associations with hyperopia with astigmatism. Further studies are needed to confirm the association of rs634990 of GJD2 with hyperopia and astigmatism. Future genetic perspectives should consider the effect of *GJD2* gene variants on gene expression levels and other regulatory mechanisms, such as methylation.

## 5. Limitation(s)

We did not measure axial length, and corneal topography was not considered in our study. However, this does not diminish the value of the study, as autorefractometer tests are also accurate and widely used to determine refraction.

## 6. Conclusions

The heritability of hyperopia and hyperopia with astigmatism was 0.654–0.492 for different eyes in patients between 20 and 40 years. Another significant *GJD2* gene, SNP rs634990, was identified, which has significant associations with the co-occurrence of hyperopia and astigmatism. Patients with the *GJD2* gene rs634990 TT genotype were found to be 2.4 times more likely to develop hyperopia with astigmatism.

## Figures and Tables

**Figure 1 genes-13-01166-f001:**
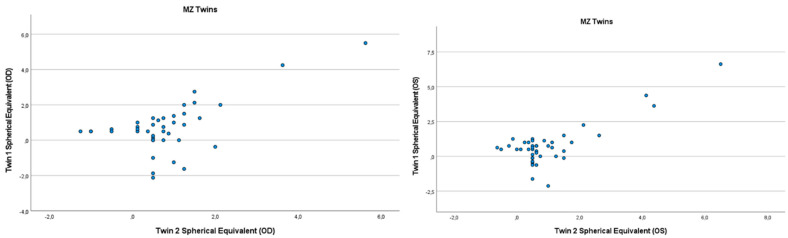
Scatter plots of hyperopia and astigmatism according to spherical equivalent for each twin right eye, for twin 1 versus twin 2 in 44 MZ twin.

**Figure 2 genes-13-01166-f002:**
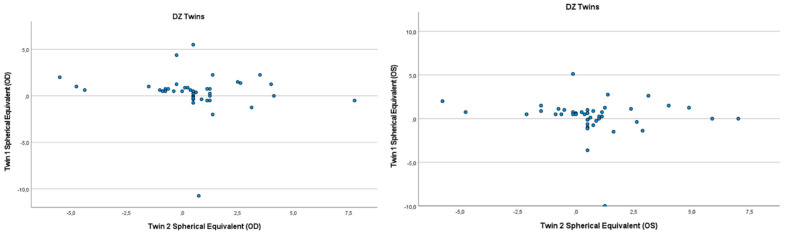
Scatter plots of hyperopia and astigmatism according to spherical equivalent for each twin right eye, for twin 1 versus twin 2 in 42 DZ twin.

**Table 1 genes-13-01166-t001:** Characteristics of twin pairs, defined by zygosity.

Characteristics	MZ Twins	DZ Twins	*p* Value
Sex, pairs			
Male	8	5	0.923 *
Female	14	10	
Male/Female		6	
Age, years			
Mean ± SE	22.18 ± 0.51	23.03 ± 0.63	0.694 **
Median	20.11	22.31	
Min, Max	18, 39	20, 40	
Spherical equivalent, D			
OD Mean ± SE	3.155 ± 0.20	2.995 ± 0.19	0.425 **
Median	3.055	2.850	
Min, Max	0.750, 6.500	0.825, 5.125	
OS Mean ± SE	3.145 ± 0.85	2.850 ± 0.63	0.071 **
Median	2.122	1.965	
Min, Max	0.725, 5.65	0.815, 3.52	

MZ—monozygotic twins; DZ—dizygotic twins, D — diopters, SE — Standard error, *p* > 0.05—comparison between MZ and DZ twins, *—*p* value for the chi-square test, **—*p* value for the Mann–Whitney test.

**Table 2 genes-13-01166-t002:** Study cohort for *GJD2* and *RASGRF1* gene polymorphisms.

	Myopia (*n* = 48)	Hyperopia (*n* = 14)	Myopia with Astigmatism (*n* = 239)	Hyperopia with Astigmatism (*n* = 72)	Control (*n* = 104)
Gender					
Male, *n* (%)	17 (35.42)	6 (42.86)	83 (34.73)	32 (44.44)	46 (44.23)
Female, *n* (%)	31 (64.58)	8 (57.14)	156 (65.27)	40 (55.56)	58 (55.77)
*p*-value *	<0.001	<0.001	<0.001	<0.001	–
Age					
Median (Min; Max)	20 (18; 40)	20 (19; 26)	20 (18; 40)	22.5 (18; 40)	20 (18; 40)
*p*-value *	0.493	0.926	0.602	0.086	–
Spherical equivalent, D					
OD					
Median	−2.75	2.35	−11.25	2.5	0.15
(Min; Max)	(−5.00; −1.5)	(1; 4.5)	(−17.35; −1.25)	(1.35; 4.37)	(−0.35; 0.35)
*p*-value *	<0.001	<0.001	<0.001	<0.001	–
OS					
Median	−2.5	2.25	−7.125	2.75	−0.125
(Min; Max)	(-3.5; −1.75)	(1.25; 3.75)	(−16.75; −1.35)	(1.875; 5.125)	(0.125; −0.375)
*p*-value *	<0.001	<0.001	<0.001	<0.001	–

* *p*-value—calculated significance level, differences are considered statistically significant when *p* < 0.05—comparison between control and refractive error group; OD—right eye; OS—left eye; D—diopters, min—minimum value; max—maximum value.

**Table 3 genes-13-01166-t003:** The associations between *GJD2* and *RASGRF1* gene variants and hyperopia with astigmatism.

Gene	SNP	Genotype	Hyperopia with Astigmatism (*n* = 72)	Control (*n* = 104)	OR	95% C.I.		*p* Value
						Lower	Upper	
*GJD2*	rs634990	CC	17	23	1.060	0.517	2.174	0.874
		CT	30	59	0.508	0.273	0.946	0.033 *
		TT	23	17	2.360	1.146	4.860	0.020 *
	rs524952	AA	17	19	1.298	0.616	2.735	0.492
		AT	31	57	0.529	0.281	0.997	0.049 *
		TT	20	17	1.863	0.888	3.907	0.100
*RASGRF1*	rs8027411	TT	16	20	1.186	0.566	2.484	0.652
		TG	45	63	1.058	0.569	1.968	0.858
		GG	11	20	0.748	0.334	1.676	0.481
	rs4778879	AA	23	41	0.671	0.352	1.276	0.224
		AG	38	39	1.792	0.959	3.348	0.068
		GG	8	16	0.656	0.263	1.632	0.364
	rs28412916	CC	22	38	0.705	0.366	1.356	0.295
		CA	38	39	1.786	0.951	3.356	0.071
		AA	8	17	0.604	0.244	1.494	0.275

*—statistically significant differences at *p* < 0.05—comparison between hyperopia with astigmatism and control; OR—odds ratio; C.I.—confidence interval.

**Table 4 genes-13-01166-t004:** Associations between haplotype for SNPs in the *GJD2* gene and hyperopia.

*GJD2* (rs634990)	*GJD2* (rs524952)	Frequency, *n* (%)		OR	95% C.I.		*p* Value
		Control	Hyperopia		Lower	Upper	
T	T	68 (76.4)	9 (69.2)	0.838	0.416	1.688	0.621
C	T	56 (62.9)	6 (46.2)	0.410	0.220	0.765	0.005 *
T	A	53 (59.6)	7 (53.8)	0.613	0.332	1.132	0.118
C	A	72 (80.9)	10 (76.9)	0.501	0.246	1.019	0.056

*—statistically significant differences at *p* < 0.05—comparison between control and hyperopia; OR—odds ratio; C.I.—confidence interval.

**Table 5 genes-13-01166-t005:** Associations between haplotype for SNPs in the *GJD2* gene and hyperopia with astigmatism.

Haplotype		Frequency, *n* (%)		OR	95% C.I.		*p* Value
*GJD2* (rs634990)	*GJD2* (rs524952)	Control	Hyperopia with Astigmatism		Lower	Upper	
T	T	68 (76.4)	49 (74.2)	0.890	0.426	1.861	0.757
C	T	56 (62.9)	26 (39.4)	0.383	0.199	0.737	0.004 *
T	A	53 (59.6)	30 (45.5)	0.566	0.297	1.077	0.083
C	A	72 (80.9)	43 (65.2)	0.441	0.212	0.918	0.029 *

*—statistically significant differences at *p* < 0.05—comparison between control and hyperopia with astigmatism; OR—odds ratio; C.I.—confidence interval.

**Table 6 genes-13-01166-t006:** Distribution of SNP rs524952 genotypes according to the degree of hyperopia.

Gene (SNP)	Genotype	Degree of Refraction, D	OD				OS			
			*p* Value	OR	95% C.I.		*p* Value	OR	95% C.I.	
					Lower	Upper			Lower	Upper
Hyperopia and Hyperopia with Astigmatism (*n* = 287)										
*GJD2* (rs524952)	AA	Low (<+2.25)	0.350	2.00	0.467	8.557	0.812	0.833	0.185	3.750
		Moderate (+2.25 to +5.00)	–	–	–	–	–	–	–	
	AT	Low (<+2.25)	0.045 *	0.255	0.067	0.971	0.768	0.848	0.284	2.533
		Moderate (+2.25 to +5.00)	–	–	–	–	0.850	1.273	0.105	15.385
	TT	Low (<+2.25)	0.151	2.625	0.702	9.809	0.592	1.400	0.409	4.791
		Moderate (+2.25 to +5.00)	–	–	–	–	0.664	1.750	0.140	21.876

*—statistically significant differences at *p* < 0.05; OR—odds ratio; C.I.—confidence interval; OD—right eye; OS—left eye; D—diopters.

## Data Availability

Data supporting the reported results are available directly from the main author upon request.

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
