# Peer review of "Association of CX36 Protein Encoding Gene GJD2 with Refractive Errors"

_genes, 2022, doi:10.3390/genes13071166_

Round 1
Reviewer 1 Report
Please extend the correlation of the GJD2 gene, GJD2 rs634990 TT genotype ,GJD2 rs634990 CT genotype, GJD2 rs524952 AT genotype, ACTAGG for 33 rs634990 and TTTAGA for rs524952 in neurological disease.
Author Response
Dear Editor and Reviewers'
We greatly appreciate the revision of our manuscript. We would like to take this opportunity to express our sincere gratitude to the reviewers and editors who helped improve the manuscript. We would also like to thank you for the opportunity to resubmit a revised manuscript. We hope that the revised manuscript will be acceptable for publication in your journal. Please also find below our point-by-point responses to the comments of the reviewers (editors). The manuscript are presented in 2 variants, one with track changes (genes-1727456_06-13.docx) and the other is clear (genes-1727456_06-13_clear.docx).
Best regards,
Edita Kunceviciene

Reviewer 2 Report
The authors aimed to evaluate the associations of the GJD2 and RASGRF1 gene polymorphisms with refractive errors and found that the heritability of hyperopia and hyperopia with astigmatism was 0.654 and 0.492 in patients between 20 and 40 years of age.
Line 61 - The hyperopia has a genetic association, revise the scientific literature.
Line 93 - Need to express the necessity of this study to the actual society. I mean why is this research important and what does it suppose for the actual knowledge of the hyperopia and hyperopia with astigmatism genetic knowledge Ent.
Line 114 - Is the refraction confirmed with any subjective method? Do you find it important to subjectively the refraction or only measured with the autorefractometer?
Line 249 - This relationship seems to be important, but what was the clinical application of this issue? What could be the future perspective on genetic modification to improve these findings?
Line 302 - A limitation and strengths section must be added at the end of the discussion.
Line 304 - A future research line section should be added at the end of the discussion.
Line 309 - The conclusion should include the daily clinical relevance of the findings in this study.
References - Where no with the MDPI instructions
References - Please update references prior to 2010
References - Include only references included on indexed journals
From line 372 to the end of the manuscript, the authors repeat al author guidelines; it denotes a low interest from the authors about the details on this manuscript.
Author Response
Dear Editor and Reviewers'
We greatly appreciate the revision of our manuscript. We would like to take this opportunity to express our sincere gratitude to the reviewers and editors who helped improve the manuscript. We would also like to thank you for the opportunity to resubmit a revised manuscript. We hope that the revised manuscript will be acceptable for publication in your journal. Please also find below our point-by-point responses to the comments of the reviewers (editors). The manuscript are presented in 2 variants, one with track changes (genes-1727456_06-13.docx) and the other is clear (genes-1727456_06-13_clear.docx).

Reviewer 3 Report
The topic of this paper is interesting. However, I would urge the authors to consider the following and revise accordingly:
i) Heritability. Would you expect (h2) differences between right and left eyes? Is the inter-eye difference significant?
ii) Please define any acronym before each citation (eg RAS), or provide list of all acronyms. Most eye care specialists will be unfamiliar with most of the stated acronyms.
iii) Last sentence in the ‘Introduction’. Is it appropriate to include this? The aim, the reason for undertaking this investigation, is not clearly defined in the ‘Introduction’.
iv) Line 116. What does ‘after soi’ mean?
v) Subject numbers. The 5 groups do not feature equal numbers of subjects. How/what procedure was used to determine the numbers of subjects required (per group) to reach statistical significance? Graphs show scatter in data and possible outliers. Are the same conclusions reached after filtering out outliers?
vi) State the conditions for exclusion and inclusion as two separate lists. Were all subjects healthy? Maybe some refractive errors were triggered by medication and/or other conditions (eg incipient cataracts, hormonal imbalances).
vii) Why was axial length not measured or considered as a means of distinguishing between myopes and hyperopes? It would be interesting to see if some of the results correlated with axial length. Why was topography not considered to distinguish between astigmats and non-astigmats?
viii) Kindly revise your reference list. Some are not directly related to the point raised in a particular sentence (eg reference 3). Delete or substitute with more appropriate ones.
Author Response

(The authors gave the same response as above.)

Round 2
Reviewer 2 Report
Coments silvestre
Author Response
Dear reviewers,
thanks for the questions, comments, and the opportunity to improve our manuscript.
Best regards,
Edita Kunceviciene
Reviewer 3 Report
I'm pleased to note the authors have considered most of the points raised. My only concerns are as follows:
a) Point vii) I can‘t find 'Limitations Section‘. Thus, I cannot verify your comment.
b) Refractive error is the product of a mismatch between several ocular parameters including, primarily, axial length and corneal topography. In terms of fundamental biology, the genetic programme that influences refractive error does so by triggering the mismatch. This should be considered in the Introduction if not the Discussion.
c) Point viii) Numbering of references have not been completely corrected in the revised manuscript (eg line 53 still refers to the references in the 1st version).
d) On second thoughts, the title of the paper is not an accurate reflection of the work the authors undertook. The current title is focused on hyperopia, yet the majority of patients were myopes. (Table 2 shows 287 cases fell in the myopic camp and 86 in the hyperopic camp). Consider modifying the title.
Author Response
Dear reviewers,
thank you very much for the opportunity to improve our manuscript. We've taken note of your comments and responded to them one by one. Answered questions and corrections are attached to the pdf file: Reviewer3_Second round.
Best regards,
Edita Kunceviciene
